# Face-Based CNN on Triangular Mesh with Arbitrary Connectivity

Hui Wang [ID], Yu Guo [ID] and Zhengyou Wang *[ID]

School of Information Science and Technology, Shijiazhuang Tiedao University, Shijiazhuang 050043, China
* Correspondence: zhengyouwang@stdu.edu.cn

**Abstract:** Applying convolutional neural networks (CNNs) to triangular meshes has always been a challenging task. Because of the complex structure of the meshes, most of the existing methods apply CNNs indirectly to them, and require complex preprocessing or transformation of the meshes. In this paper, we propose a novel face-based CNN, which can be directly applied to triangular meshes with arbitrary connectivity by defining face convolution and pooling. The proposed approach takes each face of the meshes as the basic element, similar to CNNs with pixels of 2D images. First, the intrinsic features of the faces are used as the input features of the network. Second, a sort convolution operation with adjustable convolution kernel sizes is constructed to extract the face features. Third, we design an approximately uniform pooling operation by learnable face collapse, which can be applied to the meshes with arbitrary connectivity, and we directly use its inverse operation as unpooling. Extensive experiments show that the proposed approach is comparable to, or can even outperform, state-of-the-art methods in mesh classification and mesh segmentation.

**Keywords:** geometry deep learning; face-based networks; convolutional neural networks; mesh analysis

## 1. Introduction

Extending 2D convolutional neural networks (CNNs) to 3D shapes is at the forefront of the field of computer graphics and computer vision. Unlike images represented on a regular grid of discrete values, generalizing CNNs to 3D shapes with irregular structure is very difficult.

Previous work has generalized CNNs to 3D shapes using regular representations: mapping 3D shapes to multiple 2D projections [1] or 3D voxel grids [2]. Further important work has involved the direct application of CNNs to the sparse point cloud representation [3], which has inspired a series of follow-up investigations [4,5]. However, the approach can take up a lot of memory or weaken the connectivity of 3D shapes.

In contrast, triangle meshes provide more efficient representation of 3D shapes and have arbitrary connectivity, which is useful for expressing shape details. They approximate a continuous shape via a set of 2D triangles in 3D space, and are widely used in modeling, rendering, animation and other applications. However, this arbitrary connectivity makes the meshes non-uniform, which means that generalizing CNNs to the meshes is difficult.

To the best of our knowledge, most methods use CNNs indirectly by transforming the meshes into other structures, such as manifolds [6–10] and graphs [11,12], and do not take into account the structural properties of the meshes. For example, PDMeshNet [13] is a novel method which converts the meshes to primal-dual graphs with the help of graph vertices for convolution and pooling. There are some novel networks that can directly apply CNNs to the meshes. They have in common that they all use local regular connectivity—there are four edges around an edge [14], and a face has three adjacent faces [15]. This makes them only construct kernels of a fixed size, which limits the receptive field of convolution. Recently, the SubdivNet [16] network was proposed, which subdivides the mesh to provide loop subdivision connectivity, with approximate plane properties to

facilitate pooling. However, remesh preprocessing is needed and it is difficult to apply directly to meshes with arbitrary connectivity.

Seeking to tap into the arbitrary connectivity of the watertight mesh, we propose a novel face-based CNN, which bases the convolution and pooling region completely on the arbitrary connectivity. Many experiments undertaken demonstrate that the proposed method performs very well in mesh classification and segmentation. In Figure 1, some chairs are shown that are accurately segmented with face labels by our method. The main contributions of the proposed technique are the following:

- We directly generalize CNNs to the watertight meshes with arbitrary connectivity using faces as the basic elements of the network.
- A sort convolution operation of faces is designed, which can obtain a larger receptive field, while preserving the invariance of convolution.
- The face collapse is used for pooling, which can extract multi-level and more robust features of meshes.

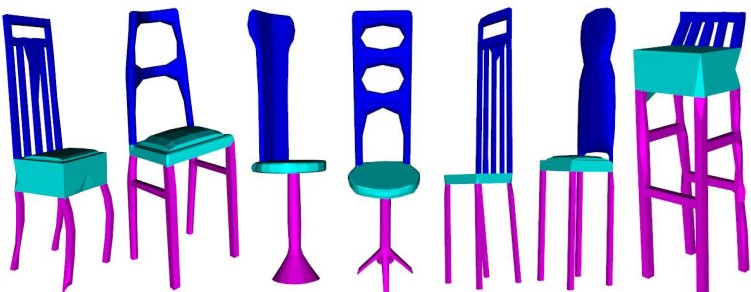

**Figure 1.** High-quality segmentation of chairs.

## 2. Related Works

There have been many previous studies of geometric deep learning, which can be classified according to their representation of geometric data [17,18]. We briefly introduce some of these studies, which are pertinent to the present investigation, and which focus on deep learning methods applied to the meshes.

### 2.1. Point Clouds

The pioneering network design based on point clouds is PointNet [3], which involves a method of direct learning on point clouds. However, it ignores the relationship between points. Inspired by CNNs, PointNet++ [4] offers an improved method, including design of a set abstraction layer to extract the local features of point clouds. PointCNN [5] and KPConv [19] improve the disorder of the point clouds and the extraction of point cloud features.

### 2.2. Voxels And Multi-Views

A regular data structure facilitates the generalization of CNNs. 3D ShapeNets [2] convert 3D shapes to 3D grids and use CNNs directly for feature extraction. Su et al. [1] projected 3D shapes onto multiple 2D images and used a view pooling layer to eliminate ordering ambiguity of input images. Qi et al. [20] conducted in-depth research to improve the above two methods. Later, studies were undertaken to explore sparse voxel representation of shapes to speed up calculation [21] and to optimize the projection perspective [22].

### 2.3. Meshes

According to the components of the backbone network, we divide deep learning methods on the meshes into the following three classifications: recurrent neural networks (RNNs), multi-layer perceptrons (MLPs) and CNNs.

### 2.3.1. RNNs

RNNs, which contain front-to-back connectivity, are good at feature extraction for sequence data. MeshWalker [23] adopts a unique approach, which involves generalizing the connectivity by random walk of vertices on mesh edges, mainly using three gated recurrent unit (GRU) layers for feature storage and extraction.

### 2.3.2. MLPs

Recently, a number of methods have been proposed which use classic MLPs as an important part of the backbone network.

HodgeNet [24] includes a learnable spectral operator which can generate features similar to classical descriptors. DiffusionNet [25], which uses spectral features as the inputs to the network, includes a diffusion layer to extract the features of vertices, and supports the input with point clouds, but requires more time for processing. These methods share the common defect that they cannot extract multi-level features.

Qiao et al. [26] proposed a method using mesh-based Laplacian encoding and pooling, which uses spectral features as input, while introducing clustering into pooling, and uses a relational network to calculate the relationship between classes. However, its flexible clustering method ignores the connectivity of the mesh faces and the structure of the meshes.

### 2.3.3. CNNs

CNNs play an important role in deep learning; however, it is very challenging to apply them to the meshes, which have a complex structure.

**Indirectly.** One approach advocates abstraction of the meshes into manifolds. Maron et al. [6] achieved convolution on the surfaces by mapping the surfaces to the planes. GWCNN [7] involves a metric alignment layer, which enables the surface features to generate a stack of 2D images to feed to the following CNNs. These methods represent the most intuitive application of CNNs based on meshes. Some methods have been proposed that use the geodesic of local patches to calculate the surface features and to design the convolution operation [8–10].

Another approach treats the meshes as graphs. Some methods regard the mesh as a special case of a graph structure, and use graph convolution to extract the vertex features [11,12]. PD-MeshNet [13] is unusual, involving the conversion of mesh faces and edges into primitive and dual graphs for convolutions, and introducing pooling of vertex merging, without excessive attention to the topology of the triangular mesh.

However, these methods overestimate the connectivity of the meshes and ignore mesh structure.

**Directly.** There are some methods which involve the direct application of CNNs to the meshes without complex preprocessing. MeshNet [15] involves a face rotation to ensure invariance of the extracted face structure features. At the same time, combined with the spatial and structural characteristics, the convolution of the mesh faces is defined using the connectivity of the faces. MeshCNN [14] uses the connectivity of edges to define the convolution of edges, and introduces a traditional mesh simplification method into the mesh pooling. These methods obtain the characteristics of rules on irregular meshes with fixed convolution kernel size; however, this limits the receptive field of convolution.

SubdivNet [16] involves the subdivision of meshes into approximate manifolds. This not only retains the mesh structure, but also improves the connectivity of the mesh. It makes the mesh itself have the characteristics of an approximate plane, so that CNNs can be easily applied to it. However, it requires complex remesh preprocessing, which substantially increases the costs of use.

Our method applies CNNs directly to the meshes with configurable convolution kernel sizes, and uses faces as basic elements in a similar way to MeshNet [15]. Mesh simplification is introduced into the pooling, using face collapse instead of edge collapse [14], so the method can work with arbitrary connectivity instead of subdivision connectivity [16].

## 3. Method

We propose a face-based CNN on triangular meshes with arbitrary connectivity, as shown in Figure 2. The core of our method has three elements: face feature, described in Section 3.2, face convolution, described in Section 3.3, and face pooling, described in Section 3.4.

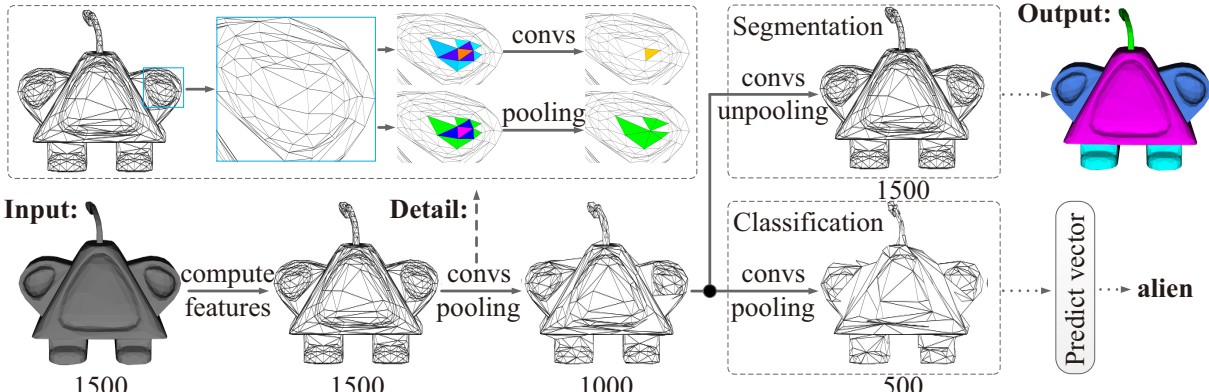

**Figure 2.** An overview of our framework for segmentation and classification, where the bottom of the model is the number of faces.

### 3.1. Notation

The mesh can be expressed as $M(V, E, F)$ with the set of vertices $V \subset \mathbb{R}^3$, edges $E \subset V \times V$ and faces $F \subset V \times V \times V$. We regard the existence of three adjacent faces in a face as the basic attribute of $F$, and, for faces with less than three adjacent faces, fill them with the face itself [15]. We express the face adjacency matrix as $G_{|F| \times 3}$, where $|F|$ is the number of faces. To resolve the ordering ambiguity, we generate $G^s$ by sorting the adjacent faces in $G$ from small to large according to the length of the common edge. For instance, in Figure 3, it can be described as:

$$[b, c, d] \rightarrow [d, b, c], \tag{1}$$

where $b, c, d$ are the indices of adjacent faces, and after sorting, the original unordered face indexes become the ordered on the right.

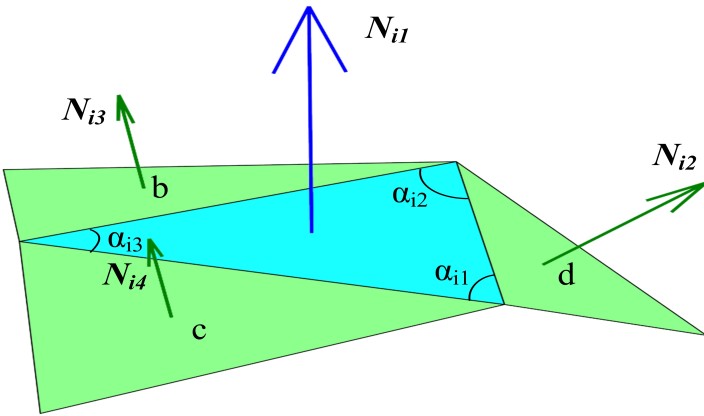

**Figure 3.** The three internal angles of a face and the normals of the face and its adjacent faces.

### 3.2. Input Features

The face feature $f_i$ ($i \in \{1, 2, \cdots, |F|\}$) is a 10-dimensional vector ($\theta_{i1}, \theta_{i2}, \cdots, \theta_{i6}, A_i, \alpha_{i1}, \alpha_{i2}, \alpha_{i3}$), including six dihedral angles, the face area and three internal angles.

**Six dihedral angles.** Given the the four normal vectors $(N_{i1}, N_{i2}, N_{i3}, N_{i4})$ of a face and its adjacent faces shown in Figure 3, we calculate six dihedral angles using the two combinations of normals according to the order of the faces in $G^s$ as:

$$
\begin{aligned}
\theta_{i1} &= \measuredangle(N_{i1}, N_{i2}), \theta_{i2} = \measuredangle(N_{i1}, N_{i3}), \\
\theta_{i3} &= \measuredangle(N_{i1}, N_{i4}), \theta_{i4} = \measuredangle(N_{i2}, N_{i3}), \\
\theta_{i5} &= \measuredangle(N_{i2}, N_{i4}), \theta_{i6} = \measuredangle(N_{i3}, N_{i4}),
\end{aligned}
\tag{2}
$$

where $\measuredangle$ is the dihedral angle between normals.

**The face area.** We normalize the area $A_i$ of each face according to the total face areas and the number of faces:

$$
A_i = A_i / (\sum_{n=1}^{|F|} A_n) \cdot |F|,
\tag{3}
$$

which makes the face area insensitive to the scale transformation of the input mesh.

**Three internal angles.** We sort the internal angles of the face, from small to large, to avoid ambiguity, as shown in Figure 3, which can be described as:

$$
[\alpha_{i1}, \alpha_{i2}, \alpha_{i3}] \rightarrow [\alpha_{i3}, \alpha_{i1}, \alpha_{i2}].
\tag{4}
$$

These features are robust to rotation, translation and scale transformation. Following the approaches of MeshCNN [14] and PD-MeshNet [13], we standardize all input features according to their mean and variance.

### 3.3. Convolution

We define a sorting convolution of faces on the meshes with a one-dimensional convolution kernel. As a consequence of the irregularity of the mesh faces, we need to set the convolution region $R_{|F| \times k}$ for each face according to the convolution kernel size $k$.

In fact, the $F$ itself and $G^s$ set a $1 \times 4$ convolution receptive field for each face. We initialize $R$ with $G^s$, then, to obtain a larger region, we continue to add adjacent faces to $R$ according to traversing $G^s$ in order and add $F$ to $R$ if $F$ is not repeated in $R$. This is stopped when the set number $k$ is reached. In Algorithm 1, we show how to obtain a larger convolution receptive field of a face in detail.

---

**Algorithm 1:** Generation of Receptive Field

---

**Input** : $|F|$: is the number of faces
$\qquad\quad$ $G^s$: face adjacency matrix
$\qquad\quad$ $k$: convolution kernel size
**Output** : $R$: convolution receptive field

Initialize: $R \leftarrow G^s$
**for** $i = 1; i \leq |F|; i++$ **do**
$\quad$ $r = 4$;
$\quad$ **while** $r \leq k$ **do**
$\quad\quad$ $j \leftarrow$ from $R_i$ in order;
$\quad\quad$ **for** $m = 1; m \leq 3; m++$ **do**
$\quad\quad\quad$ $t = G^s{}_{jm}$;
$\quad\quad\quad$ **if** $t$ *not in* $R_i$ **then**
$\quad\quad\quad\quad$ $R_{ir} = t$ and $r++$;
$\quad\quad\quad$ **end if**
$\quad\quad$ **end for**
$\quad$ **end while**
**end for**

---

The receptive field for different $k$ is shown in Figure 4; it can be seen that $k = 4$ and $k = 10$ are similar to the one ring and two ring receptive fields of 2D convolution. This construction method for the convolution receptive field can be extended to arbitrary polygonal meshes. The convolution result $\boldsymbol{f}_i$ is given by:

$$\boldsymbol{f}_i = \sum_{n=1}^{k} \delta_n \cdot \boldsymbol{f_i}^n, \tag{5}$$

where $\delta_n$ is the $n$th learnable parameter in a convolution kernel, and $\boldsymbol{f_i}^n$ is the feature of the $n$th face in $\boldsymbol{R}_i$. The $\boldsymbol{f}_i$ are sorted in $\boldsymbol{R}_i$. No matter how the meshes rotate, $\boldsymbol{R_i}$ will not change. So this convolution operation is rotation invariant.

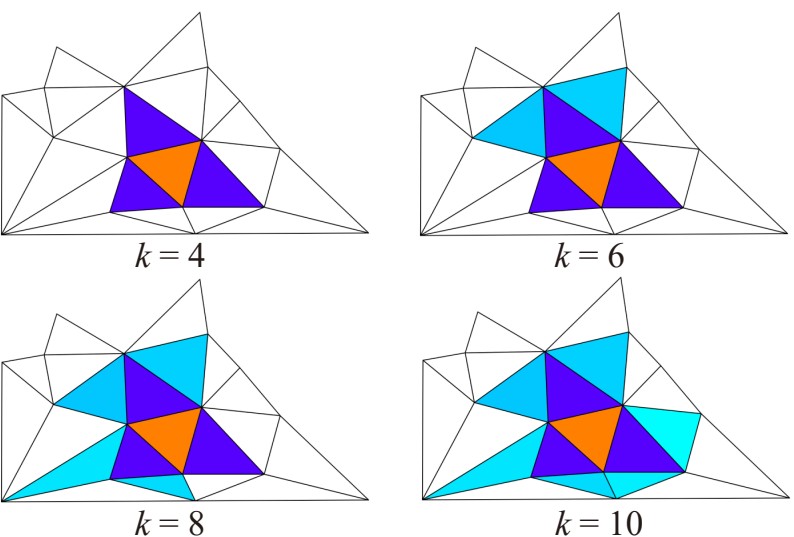

**Figure 4.** Visualization of the convolution receptive fields of different kernel size $k$. The orange parts are the center faces, which, together with the blue faces, constitute the convolutional receptive fields of the center faces.

### 3.4. Pooling

We take the face collapse as the face pooling, which can be applied to the meshes with arbitrary connectivity. As shown in Figure 5, we set the collection of colored faces as the pooling region; it is evident that the face pooling depends only on the basic attributes of the mesh face.

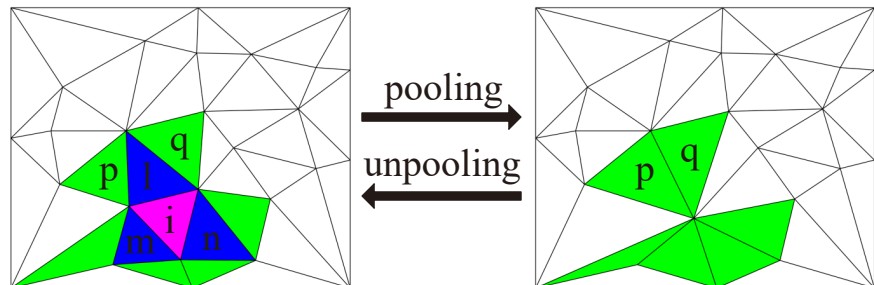

**Figure 5.** Pooling and unpooling. The purple part is the center face, which, together with the blue faces, makes up the pooling faces. Similar to the mesh simplification of face collapse, after pooling, they will disappear.

First, we select disjoint pooling regions in order according to the face weight $w$. With reference to the tradition mesh simplification methods, we compute $w$ as the difference between the features of the faces and the adjacent faces. The calculation method of each face weight $w_i$ is given by:

$$w_i = \sum_{n=1}^{3} ||f_i^1 - f_i^{n+1}||^2. \tag{6}$$

Then the faces are collapsed for pooling in sequence, which are arranged in ascending order by $w$. It should be noted that the pooling changes the structure of the meshes; therefore, we need to update $\boldsymbol{G}^s$ at the same time.

We use the number of faces as the stop criterion, and aggregate the features according to the connectivity of the faces. In Figure 5, the aggregation of features can be defined as:

$$\begin{aligned} f_p &= avg(f_p, f_i, f_l), \\ f_q &= avg(f_q, f_i, f_l), \end{aligned} \tag{7}$$

where $i, l, m, n, p, q$ are the indices of the corresponding faces marked.

Finally, as a result of the reduction in the number of faces, it is necessary to index them and update $\boldsymbol{G}^s$ again. If $k$ of the next layer is more than four, we need to reconstruct $\boldsymbol{R}$. The collapse of faces may produce non-manifold faces, which we allow. However, non-manifold faces, and their surrounding faces, cannot participate in the next pooling.

### 3.5. Unpooling

For face unpooling, which is the inverse operation of pooling, we can restore the features of each face from the aggregated features. We obtain the $\boldsymbol{R}$, which is saved before pooling, for the next convolution operation. It should be noted that the unpooling operation, and its corresponding pooling operation, which we designed, are reversible.

During pooling, we save the feature mapping matrix, which maps the original face features before pooling, to the new face features after pooling. We can directly use its inverse matrix to restore the higher resolution features. Similar to the aggregation of features in pooling, as shown in Figure 5, the diffusion of aggregated features can be defined as:

$$\begin{aligned} f_l &= avg(f_p, f_q), \\ f_i &= avg(f_l, f_m, f_n), \end{aligned} \tag{8}$$

where $f_m$ and $f_n$ are calculated in the same way as $f_l$, which are the mean features of its two adjacent green faces.

### 3.6. Network Architecture

Our proposed face convolution and pooling are similar to the edge convolution and pooling of MeshCNN [14], which can directly apply classical 2D CNNs to the meshes, such as ResNet [27] or U-Net [28]. As shown in Figure 6, our network architecture only makes a slight modification to MeshCNN [14], by reducing the number of convolution layers and pooling by half. We set $k = 10$ for each convolution layer.

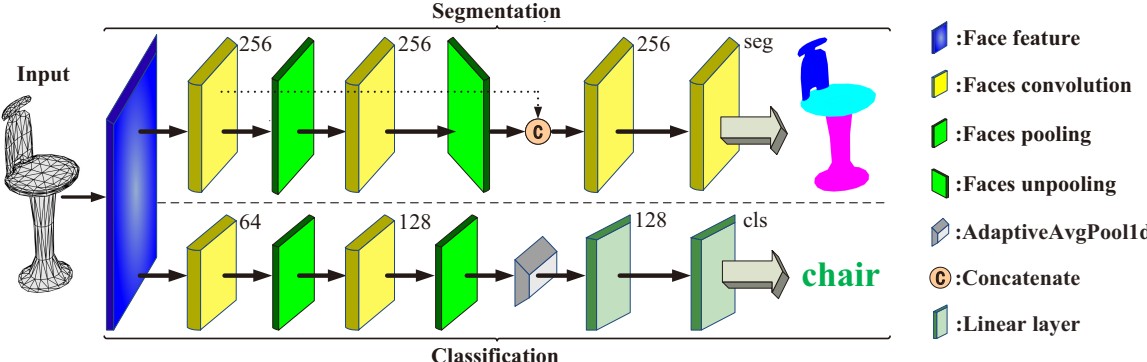

**Figure 6.** The architecture of the network for segmentation and classification, where the top of the block is the number of output channels for the features.

For segmentation, each face convolution has three residual convolutions and the last two face convolutions have an additional convolution layer to reduce the number of feature channels. The convolution parameters are [256, 256, 256, seg]. The pooling parameters are [1000, 700] or [1500, 1000], depending on the number of faces. For classification, each face convolution has a residual convolution. The convolution, MLP and pooling parameters are [64, 128], [128, cls] and [500, 400, 300]. Obviously, we only use 18 and 4 convolution layers for segmentation and classification. The above parameters are the numbers of output channels and face numbers after pooling. The seg and cls are the numbers of segmentation and classification. For the other details, we use the same configurations as MeshCNN [14].

## 4. Experiment

We tested the performance of our method on two basic tasks: mesh classification and mesh segmentation.

### 4.1. Implementation Details

For each dataset, we trained our network with a batch size of 16 models with 200 epochs for classification and 600 epochs for segmentation. Cross-entropy loss was used for the above two tasks. The initial learning rate was 0.001. After 100 epochs, the learning rate decreased linearly and finally dropped to 0.

**Augmentation.** Firstly, we used the same scale transformation based on a normal distribution as MeshCNN [14] with $\mu = 1$ and $\sigma = 0.1$ to scale the meshes randomly. Then, we picked forty percent of the edges, and set the disturbance coefficient $d = 2$. For example, we randomly selected a vertex on the edge and let it randomly disturb in the limited region. The specific formula for calculating the disturbance distance *len* is as follows:

$$len = d \cdot L \cdot (\theta/10), \tag{9}$$

where $L$ and $\theta$ are the length and dihedral angle of the selected edge. Then, we mark a random number between $-len$ and $len$ as $r$. So the random disturbance of vertices $(x, y, z)$ can be given by:

$$(x, y, z) = (x, y, z) + (r^1, r^2, r^3), \tag{10}$$

where $r^1$, $r^2$ and $r^3$ represent three different random numbers. Following the practice of MeshCNN [14], we generated 20 augmented versions of each mesh.

### 4.2. Classification

We tested our method on the simplified meshes provided by MeshCNN [14]. Each mesh contains approximately 500 faces. As a matter of routine, the accuracy is the ratio of correctly predicted meshes.

SHREC11 [29] contains 30 classes of models. There are 20 models in each class, making a total of 600 models. We follow the setup in MeshCNN [14], where the dataset contains two different splits: 16 training and 4 tests in each class (Split 16) and 10 training and 10 tests in each class (Split 10). The training set model of the two evaluation methods was randomly generated, and the final accuracy was the average value of three random times.

Cube engraving [14] was modeled as a set of cubes with shallow icon engravings. There were 4381 models in 22 classes, with 3722/659 in the train/test split.

Compared with other recently proposed methods listed in Table 1, our approach proved very competitive. Unlike the remeshing preprocessing of SubdivNet [16], our method can be directly applied to meshes with arbitrary connectivity.

**Table 1.** Classification results on SHREC11 and Cube engraving.

| Method | Split 16 | Split 10 | Cube |
| --- | --- | --- | --- |
| GWCNN [7] | 96.6 | 90.3 | - |
| MeshCNN [14] | 98.6 | 91.0 | 92.2 |
| HSN [10] | - | 96.1 | - |
| PD-MeshNet [13] | 99.7 | 99.1 | 94.4 |
| MeshWalker [23] | 98.6 | 97.1 | 98.6 |
| LaplacianNet [26] | 98.0 | 90.3 | - |
| HodgeNet [24] | 99.2 | 94.7 | - |
| DiffusionNet [25] | - | 99.7 | - |
| SubdivNet [16] | **100** | **100** | **100** |
| Ours | **100** | **100** | 99.4 |

### 4.3. Segmentation

We tested our method on the simplified meshes provided by MeshCNN [14] and PD-MeshNet [13]. Each mesh contained 1000 or 1500 faces. As is common, the accuracy was defined as the ratio of correctly predicted mesh elements.

**Metrics.** Different methods extract distinct mesh element features, so various labels are used for training and testing, which can be roughly divided into three categories: points [3,4], edges [14] and faces [13]. The final accuracy of MeshCNN [14] involves calculation of the area weights of the edges; PD-MeshNet [13] applies evaluation criteria based on the length of each edge. HodgeNet [24] also represents the accuracy based on area weights of the faces. Inspired by HodgeNet [24], we can convert the features of points, edges and faces to each other by means of average pooling or maximum pooling. In order to provide a fair comparison with other methods, we did not change the existing evaluation standards, but directly used the labels of the public data for training and evaluation.

**Verification.** For mesh segmentation, we chose two ways to verify the effectiveness of our method. One was to use the labels of faces for training and testing, which was the same as for PD-MeshNet [13]. The other was to use the labels of edges for training and testing, as with MeshCNN [14]. The output of our network needs to convert the features of faces into the features of edges through average pooling.

COSEG [30] contains three types of models: aliens, vases and chairs. There are 200, 300 and 400 models in each category, and each model is divided into three or four parts. Each category was divided into 85% for training and 15% for testing. We used the hard labels of the edges to visualize and make a qualitative comparison with MeshCNN [14], as shown in Figure 7. We report the accuracy for some recent methods in Tables 2 and 3. Our method was found to have an accuracy close to, or even better than, the state-of-the-art methods.

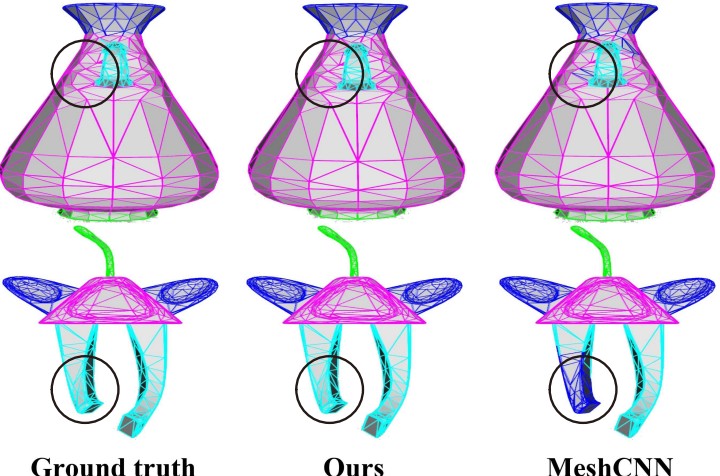

**Ground truth**      **Ours**      **MeshCNN**

**Figure 7.** Comparison of qualitative segmentation results with MeshCNN [14] on a vase (**top**) and an alien (**bottom**) of the COSEG dataset.

**Table 2.** Segmentation results on COSEG with faces labels.

| Method | Vases | Chairs | Aliens |
|---|---|---|---|
| MeshCNN [14] | 92.4 | 93.0 | 96.3 |
| PD-MeshNet [13] | 95.4 | 97.2 | **98.1** |
| HodgeNet [24] | 90.3 | 95.7 | 96.0 |
| Ours | **95.9** | **99.2** | 97.8 |

**Table 3.** Segmentation results on COSEG with edges labels.

| Method | Vases | Chairs | Aliens |
|---|---|---|---|
| PointNet++ [4] | 94.7 | 98.9 | 79.1 |
| PointCNN [5] | 96.4 | 99.3 | 97.4 |
| MeshCNN [14] | 97.3 | 99.6 | 97.6 |
| MeshWalker [23] | **98.8** | 99.6 | 99.1 |
| SubdivNet [16] | 98.1 | 99.5 | **99.4** |
| Ours | 98.4 | **99.8** | 98.7 |

The human body segmentation [6] dataset contains 399 models, 381 models for training and 18 models for testing. The accuracy results are shown in Table 4.

We visualize some qualitative comparisons with PD-MeshNet [13] and HodgeNet [24] in Figure 8, which shows that our approach is competitive.

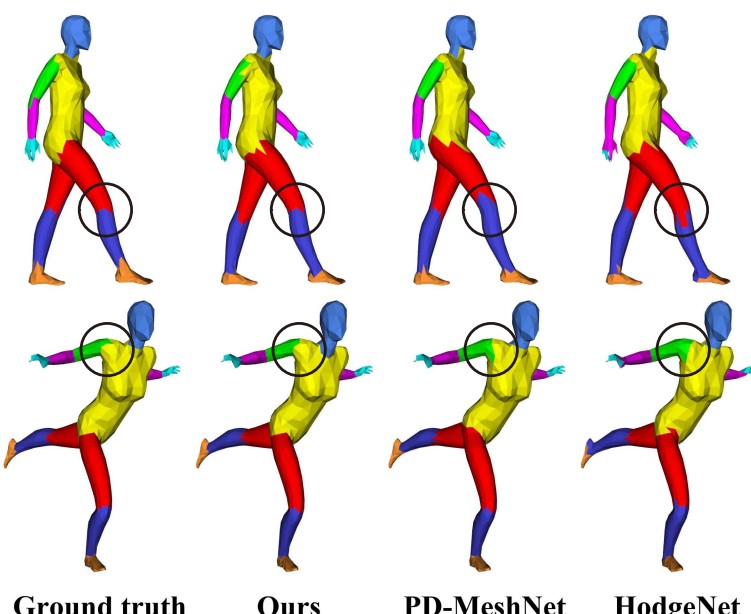

**Ground truth**　　　**Ours**　　　**PD-MeshNet**　　　**HodgeNet**

**Figure 8.** Our method can achieve more accurate boundaries for a human body dataset.

**Table 4.** Segmentation results on human body segmentation with different labels.

| Method | Edges | Faces |
|---|---|---|
| PointNet++ [4] | 90.8 | - |
| MeshCNN [14] | 92.3 | 85.4 |
| PD-MeshNet [13] | - | 85.6 |
| HodgeNet [24] | - | 85.0 |
| MeshWalker [23] | 94.8 | - |
| DiffusionNet [25] | 95.5 | 90.8 |
| SubdivNet [16] | **96.6** | **91.7** |
| Ours | 93.1 | 87.4 |

It should be noted that, in contrast to other methods, SubdivNet [16], which involves remesh preprocessing, cannot use simplified labels for training, but directly projects the prediction results of the high resolution meshes onto the simplified meshes. It uses different train-test splits in vases and chairs. However, our method can be applied to the meshes with arbitrary connectivity, so that it can be easily compared with other methods.

### 4.4. Ablation Study

We conducted some ablation studies to prove the effectiveness of our method and the rationality of selecting parameters on the dataset of COSEG vases [30].

**Input features.** Table 5 shows the necessity of the selected input features. When some input features are removed, the segmentation accuracy degrades.

**Table 5.** Ablation experiments of the input features on the COSEG vases for segmentation.

| Input | Accuracy |
| --- | --- |
| No dihedral angles | 94.32 |
| No area | 94.85 |
| No internal angles | 95.59 |
| Full | 95.90 |

**Convolution kernel sizes.** In Table 6, we compare performances with different convolution kernel sizes. When $k$ is 10, the segmentation accuracy is the highest.

**Table 6.** Ablation experiments for convolution kernel size on the COSEG vases for segmentation.

| Kernel Size | Accuracy |
| --- | --- |
| 4 | 94.31 |
| 10 | 95.90 |
| 20 | 95.08 |
| 30 | 94.10 |

**Pooling strategies.** We represent some unique pooling strategies of meshes in Table 7. For the selection of pooled regions, it is better to select faces with the smallest difference in features from the adjacent faces, rather than faces with the smallest features, which is different from MeshCNN [14].

**Table 7.** Ablation experiments for pooling evaluated on COSEG vases segmentation.

| Collapse | Strategy | Accuracy |
| --- | --- | --- |
| Face | MeshCNN-like | 94.64 |
| Face | No limit | 95.10 |
| Face | Limit | 95.90 |
| Edge | Limit | 95.51 |

We also found that limiting pooling, which prevents selected pooling regions from overlapping, can improve the performance of the network. Non-overlap means that, in a pooling layer, faces that have been selected as pooling regions can no longer be selected, and ensures that the selected pooling regions are uniform, as far as possible, similar to the pooling of SubdivNet [16]. MeshCNN-like means the faces with the smallest features for pooling are selected, without considering whether the pooling regions are overlapping.

We sought to introduce edge collapse to replace the face collapse of the pooling. However, it is necessary to not only maintain the corresponding relationship between the edges and faces, but also to maintain the index of edges. The implementation is complex and time-consuming.

*4.5. Efficiency*

We compared the average training time for each mesh, the network parameters and the GPU memory usage of our network with some methods based on the other mesh elements. Table 8 shows that our face-based method was more efficient than edge-based MeshCNN [14] and point-based HodgeNet [24], which are not optimized in parallel. HodgeNet [24] runs on CPU, so we do not give GPU memory usage. We measured all items on a 4GB GEFORECE GTX GPU and an 8-core CPU with a batch size of one mesh; the final results were the average of 300 samples on SHREC11 [29].

**Table 8.** Time and space complexity for SHREC11 Split 10.

| Method | Time (ms) | Params (M) | GPU (M) |
|---|---|---|---|
| MeshCNN [14] | 113 | 1.323 | 945 |
| HodgeNet [24] | 971 | 0.053 | - |
| Ours | 56 | 0.312 | 875 |

HodgeNet [24], which generalizes a simple complex to mesh-based deep learning, can obtain good results with a small number of parameters. However, it is difficult to use the GPU to accelerate its operation by relying only on existing deep learning tools, which are more suitable for matrix processing rather than geometry processing. In contrast, our method generalizes CNNs to the meshes, which is very efficient with the help of existing CNNs.

## 5. Conclusions

This paper introduces a novel deep learning method involving the direct application of CNNs to the watertight meshes with arbitrary connectivity. The face is used as the basic element for feature extraction, convolution and pooling. Sorting is used to deal with the irregularity of the meshes and the order ambiguity of face convolution, and learnable face simplification is used to extract multi-level features. Classification and segmentation experiments undertaken demonstrated its effectiveness and that its simple implementation enabled straightforward application to various mesh analysis tasks.

**Future work.** Non-watertightness and high resolution of the mesh are two obstacles to the application of our method. A common solution is to make the meshes watertight and to simplify them, referencing MeshCNN [14], and then to perform various mesh analysis tasks. An avenue for future research is to explore post-processing methods to simplify the meshes, such as the application of boundary smoothing to mesh segmentation [23]. In addition, how to improve different mesh analysis tasks, using correspondingly more advanced network architectures, is a potential area for further research.

**Author Contributions:** Conceptualization, H.W.; methodology, H.W., Y.G. and Z.W.; validation, H.W., Y.G. and Z.W.; writing—original draft preparation, Y.G.; writing—review and editing, H.W. and Z.W. All authors have read and agreed to the published version of the manuscript.

**Funding:** This work was jointly supported by the National Natural Science Foundation of China under Grants No. 61972267, the Hebei Education Department Foundation for the Cultivation of Innovative Ability of Postgraduate Students under Grants No. CXZZSS2022105, the Nature Science Foundation of Hebei Province under Grants No. F2019210306 and the Key Project of Science and Technology Research of Hebei Province University under Grants No. ZD2021333.

**Institutional Review Board Statement:** Not applicable.

**Informed Consent Statement:** Not applicable.

**Data Availability Statement:** Publicly available datasets were provided by MeshCNN [14] and PD-MeshNet [13] in this paper. The data can be found from the github of the above references: [https://ranahanocka.github.io/MeshCNN] and [https://github.com/MIT-SPARK/PD-MeshNet] (accessed on 4 August 2022).

**Conflicts of Interest:** The authors declare no conflict of interest.

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
