# Peer review of "Face-Based CNN on Triangular Mesh with Arbitrary Connectivity"

_electronics, doi:10.3390/electronics11152466_

Round 1

Reviewer 1 Report

This paper designs convolutional neural networks for mesh data learning. The proposal is novel, interesting, and of practical significance. I suggest publication after minor changes to correct typos like "PDMeshNet proposes a novel method, which convert the mesh". 

[Summary] This manuscript presents a new deep learning method which directly applies CNNs to the meshes with arbitrary connectivity. The face is used as the basic element for feature extraction, convolution and pooling. Sorting is utilized to deal with the irregularity of the meshes and the order ambiguity of face convolution, and learnable face simplification is used to extract multi-level features. Experiments of classification and segmentation show its effectiveness and efficiency.

[Overall evaluation] This pap manuscript is overall well motivated and well written. The proposal is novel, interesting, and of practical significance. I suggest publication after minor changes with the following comments and suggestions.

[Comments] Comment 1: The authors are suggested to go through a careful proofreading to correct the following typos and grammar mistakes:

1.1) “PDMeshNet proposes a novel method, which convert the mesh”

1.2) “We follow the setup in MeshCNN [14], two evaluation methods are adopted”

1.3) In Equation (1), the symbol “-->=” should be explained.

Comment 2: The proposed algorithm seems a bit involved, and it would be helpful to the readers if the authors provide a demo of the source code (once the manuscript is accepted for publication).

Comment 3: The proposed method is based on calculating simple geometrical features. Another direction to deal with mesh data is using simplicial complexes. It is interesting to see some discussions on difference between the proposed models and simplicial-type neural networks.

Reviewer 2 Report

The authors proposed a novel face-based CNN model on triangular mesh. The results look promising, but the paper is unfortunately not well-written and not organized. I hope authors could extensively revise this paper to improve its quality.

1.     Authors should pay attention to the writing of the paper. There are multiples typos such as: “More powerful works is directly apply…”

2.     The introduction of this paper needs to be revised extensively so others can really understand authors’ motivation and background of this research.  

3.     There is no explanation of notations in some equations, such as Eq. (1), what are the meaning of “b,c,d”?  Authors should explain.

4.     There is no explanation in the legend of figures, such as in Figure (6), what are top figures and bottom figures?

5.     Figure 5 is still not clear to show how the pooling and unpooling works. What are the “l,I,m,n”? In Eq. (8), there are only “l” and “i”, so after the unpooling, where are “m,n”?

6.     In Figure 4, what are the blue and orange parts?

7.     The architecture of this model is still not clear.  What are the components of this model and how this model is trained?

Round 2

Reviewer 2 Report

The paper's quality has been improved after the revision.